# Probing multi-spinon excitations outside of the two-spinon continuum in the antiferromagnetic spin chain cuprate $Sr_2CuO_3$

J. Schlappa[1,2], U. Kumar [3], K.J. Zhou [2,4], S. Singh[5], M. Mourigal [6,8], V.N. Strocov[2], A. Revcolevschi[7], L. Patthey[2], H.M. Rønnow [6], S. Johnston [3] & T. Schmitt[2]

One-dimensional (1D) magnetic insulators have attracted significant interest as a platform for studying quasiparticle fractionalization, quantum criticality, and emergent phenomena. The spin-1/2 Heisenberg chain with antiferromagnetic nearest neighbour interactions is an important reference system; its elementary magnetic excitations are spin-1/2 quasiparticles called spinons that are created in even numbers. However, while the excitation continuum associated with two-spinon states is routinely observed, the study of four-spinon and higher multi-spinon states is an open area of research. Here we show that four-spinon excitations can be accessed directly in $Sr_2CuO_3$ using resonant inelastic x-ray scattering (RIXS) in a region of phase space clearly separated from the two-spinon continuum. Our finding is made possible by the fundamental differences in the correlation function probed by RIXS in comparison to other probes. This advance holds promise as a tool in the search for novel quantum states and quantum spin liquids.

[1] European X-Ray Free-Electron Laser Facility GmbH, Holzkoppel 4, 22869 Schenefeld, Germany. [2] Photon Science Division, Paul Scherrer Institut, 5232 Villigen PSI, Switzerland. [3] Department of Physics and Astronomy, The University of Tennessee, Knoxville, TN 37996, USA. [4] Diamond Light Source, Harwell Science and Innovation Campus, Didcot, Oxfordshire OX11 0DE, UK. [5] Department of Physics, Indian Institute of Science Education and Research, Dr. Homi Bhabha Road, Pune 411008, India. [6] École Polytechnique Fédérale de Lausanne, 1015 Lausanne, Switzerland. [7] Institut de Chimie Moléculaire et des Matériaux d'Orsay, Université Paris-Sud 11, UMR 8182, 91405 Orsay, France. [8] Present address: School of Physics, Georgia Institute of Technology, Atlanta, GA 30332, USA. Correspondence and requests for materials should be addressed to J.S. (email: justine.schlappa@xfel.eu) or to S.J. (email: sjohn145@utk.edu) or to T.S. (email: thorsten.schmitt@psi.ch)

When confined to one spatial dimension (1D), systems of interacting electrons host an assortment of macroscopic many-body phenomena, including quantum critical magnetic states with collective excitations carrying fractional quantum numbers. For this reason, quasi-1D magnetic insulators have attracted wide experimental and theoretical interest as an ideal playground for studying quantum many-body phenomena. Owing to numerous experimental realizations of such models in real materials, some of the most stringent tests of quantum many-body theory have been conducted in 1D[1–19].

The 1D Heisenberg antiferromagnet (HAFM), where localized spins $S$ interact with their nearest neighbours via an exchange interaction $J$, is perhaps the simplest and best understood of these systems; the spin-1/2 case is an important reference system that can be solved exactly using the Bethe ansatz. The ground state is a macroscopic SU(2)-symmetric singlet, in which quantum fluctuations suppress long-range order, leading to a spin liquid ground state even in the limit of zero temperature. The elementary excitations are collective spin density fluctuations called spinons, which are fractional excitations carrying spin ½ but no charge. Spinons generated in 1D HAFM through an elementary spin-flip process, for example, during inelastic neutron scattering (INS) or resonant inelastic x-ray scattering (RIXS), are created in pairs. As such, the low-energy magnetic excitations are spanned by states involving an even number of spinons forming manifolds of two-, four-, six-spinon … continua and so forth.

The magnetic excitation spectrum has been observed for different realizations of the 1D HAFM by INS[1,2,5,6,8] and by RIXS[4,10,20–22]. The spectral weight captured by these studies, assigned to the triplet manifold, is located entirely within the boundaries of the two-spinon continuum. The reason for this is now well understood through applications of analytical theory[23,24] or numerical approaches like density matrix renormalization group[12,13]. While the allowed phase space for four-spinon excitations (and greater) is much larger than for two-spinon excitations[22,24], kinematic constraints on the matrix elements between the spinon manifolds lead a situation where the multi-spinon states only contribute significantly for momentum and energy transfers within the boundaries of the two-spinon continuum. This picture has been confirmed by detailed comparisons between INS experiments[5,12] and exact calculations of the dynamical structure factor (DSF)[5,23,24], which find the two-spinon excitations account only for ~73–74% of the total detected spectral weight, while four-spinon excitations exhaust the majority of the remaining sum rule.

While the exact solution of the pure HAFM model predicts that DSF has a small amount of spectral weight located between the upper boundary of the two-spinon continuum and the upper boundary of the four-spinon continuum[24], such a small signal has yet to be detected. Four-spinon excitations have been reported outside the two-spinon continuum in the metallic $4f$ electron material $Yb_2Pb_2Pb$[25] and the 1D ferromagnet $LiCuVO_4$[16–18]. Both materials, however, have physics beyond the simple HAFM such as long-range hopping in $Yb_2Pb_2Pb$ or frustration in $LiCuVO_4$. A direct observation of higher-order spinon excitations separated from the two-spinon continuum in the prototypical case of a 1D HAFM with nearest-neighbour interactions only is still lacking. Here, we show that RIXS at the O $K$-edge allows for such an observation, a capability that results from the fundamentally different correlation function that it probes compared to, for example, the spin correlation function of the DSF[22,26].

RIXS is a photon-in photon-out spectroscopy technique where photons inelastically scatter from a sample[20]. In a RIXS experiment, the photon energy $\hbar\omega_{in}$ of the incident x-rays is tuned close to an absorption edge of an atomic species, thereby initiating an electron transition between a core level and an unoccupied

valence-band state. This process creates an intermediate state with an additional electron either in the valence or conduction band and a hole in the core level. This core-hole excited state will decay on a femtosecond timescale, leaving the system in a long-lived valence-band excited state. Since x-ray photons carry substantial momentum (in contrast to the light of optical or vacuum ultraviolet wavelengths), the triggered valence-band excitations can be studied both in the energy and the momentum domain. Thus, RIXS can be viewed as momentum-resolved resonant Raman spectroscopy, suitable for mapping dispersions of excitations in quantum materials. The RIXS selection rules allow studies of magnetic excitations with $\Delta S_{tot} = 0$ (where $S_{tot}$ is the total spin of the system) and—in case of a strong spin–orbit coupling in the initial, intermediate, or final state—$\Delta S_{tot} = 1$.

RIXS has been used to probe electronic excitations involving charge[4,19,20], orbital[4,10,19], spin[4,10,14,27–30], and lattice[15,31] degrees of freedom in a wide range of materials. Studies on the dynamic magnetism have largely focused on cuprates, where $\Delta S_{tot} = 1$ direct spin-flip excitations can be investigated at the Cu $L_3$-edge[32]. Indeed, in Cu $L_3$ RIXS measurements of the quasi-1D spin-chain cuprate $Sr_2CuO_3$, two-spinon continuum excitations could be probed (with indications of also four-spinon excitations)[4]. Studies in other cuprate materials revealed two-triplon excitations in the spin-ladder system $Sr_{14}Cu_{24}O_{41}$[14] and magnon excitations in many quasi two-dimensional superconducting cuprates[27,28,31,32].

In this article, we report momentum-resolved oxygen $K$-edge RIXS studies of the quasi-1D spin-chain cuprate $Sr_2CuO_3$, one of the best realizations of the 1D HAFM. We observe magnetic excitations in two non-overlapping regions of phase space. Through detailed modeling within the $t-J$ model, we show that one set of these excitations is quite similar to triplet excitations generally associated with DSF, while the other set corresponds to predominantly four-spinon excitations. Specifically, four-spinon excitations centered at 500 meV energy transfer give a strong and broad response around the $\Gamma$-point ($q = 0$, where $q$ is the momentum transfer along the chain) that is well separated from the boundaries of the two-spinon continuum. Our results constitute the discovery of a channel for the creation of magnetic excitations in 1D materials, beyond those resulting from elementary spin-flip excitations. We argue that this capability stems from the spin and charge dynamics of the intermediate state, which grants access to fundamentally different correlation functions, not captured by a simple two-site correlation function.

## Results

**Experimental results.** The low-energy electronic degrees of freedom in the charge-transfer insulator $Sr_2CuO_3$ are formed from the $CuO_4$ plaquettes, which are arranged into 1D corner-shared chains[4,33], as shown in Fig. 1a. In the atomic limit, the Cu ion is in a $d^9$ valence state, with a single hole occupying the Cu $3d_{x^2-y^2}$ orbital. There is, however, significant hybridization between the Cu $3d$ and $2p$ orbitals of the surrounding oxygen, resulting in a substantial isotropic superexchange interaction $J \sim 250$ meV[4,7,8] between the Cu spins. In the real material, the individual -Cu-O-Cu- chains are weakly coupled such that the system has a bulk Néel temperature of $T_N \sim 5$ K[34]. Above this temperature the chains decouple and become nearly ideal realizations of the 1D HAFM, as evidenced by the observation of the two-spinon continuum in INS[8] and Cu $L_3$ RIXS[4]. The latter RIXS study also found evidence for spin–orbit separation effects in $Sr_2CuO_3$, further underscoring the importance of the 1D physics.

Figure 1b shows the x-ray absorption (XAS) data of $Sr_2CuO_3$ measured at the O $K$-edge (a $1s \rightarrow 2p$ resonance). The intensity reflects the partial density of the unoccupied valence and

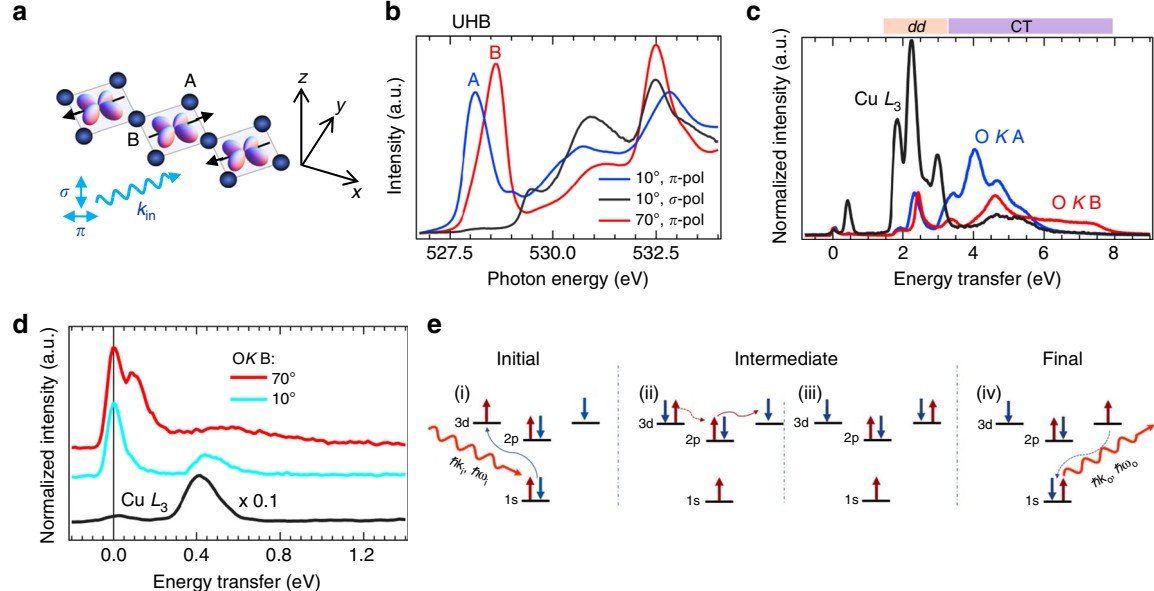

**Fig. 1** Summary of the experimental data at the oxygen $K$-edge. **a** Cartoon sketch of the Cu-O-Cu corner-shared chains forming the active low-energy degrees of freedom in $Sr_2CuO_3$, and of the incident-light geometry. The Cu atoms are primarily in a $d^9$ valence state, where a single hole occupies each of the Cu $3d_{x^2-y^2}$ orbitals and interacts antiferromagnetically with its in-chain neighbours. **b** The polarization dependence of the XAS spectra. $\sigma$-polarized light (black solid line) probes unoccupied states perpendicular to the $CuO_4$-plaquettes, having no spectral weight at the UHB (there are no apical oxygens). Data obtained with $\pi$-polarized light at incidence angles of 10° (grazing incidence geometry, blue line) and 70° (close to normal incidence, red line) primarily probes the out-of-chain (A) and in-chain (B) oxygen sites, respectively. The difference in the pre-peak resonance corresponds to the differences in the chemical environments of these two oxygen sites (chemical shifts), where the B site hosts the plaquette-connecting oxygen orbital[33]. **c** Polarization dependence for $\pi$-polarized O $K$-edge RIXS data for incident energies tuned to the A (blue) and B (red line) peaks in the XAS shown in **b**. Incident angles as in **b**: 70° (10°) corresponds to $q \approx 0$ ($q \approx \pi/2a$). The $\pi$-polarized Cu $L_3$-edge RIXS data at 20° incidence angle ($q \approx \pi/2a$) is also shown for comparison (black line). The RIXS spectra are normalized to acquisition time. The peaks above 1.8 eV energy transfer are associated to $dd$ (orbiton) and charge transfer (CT) excitations, as indicated. The peak below 0.6 eV in the Cu $L_3$ data corresponds to multi-spinon excitations[4]. **d** The Cu $L_3$ and O $K$ B-resonance RIXS data from **c** and B-resonance for 10° incidence angle ($q \approx \pi/2a$, turquoise line), now focusing on the first 1.3 eV in energy transfer, where several low-energy spin excitations are found. The RIXS data are reported in arbitrary units (a.u.). **e** Sketch of the double spin-flip process across two Cu sites at the oxygen $K$-edge

conduction band states, here projected onto the oxygen orbitals. We observe a sharp excitonic structure in the pre-edge region and broad continuum states at energies above 529 eV. The excitonic peak corresponds to excitations of the O 1$s$ core electron into the upper Hubbard band (UHB), creating a Cu $3d^{10}$ state[33]. This excitation is allowed by the sizable hybridization between the O 2$p$ and Cu 3$d$ orbitals. The UHB XAS peak depends strongly on the polarization of the incident photons reflecting the strong structural and electronic anisotropy of the system[33]. In particular, the suppression of intensity for $\sigma$-polarized light indicates that the unoccupied states are oriented in the plane of the $CuO_4$ plaquettes, whereas the energy shift upon changing the incidence angle to $\pi$-polarized light reflects differences in coordination between the out-of-chain and the in-chain oxygens (indicated in Fig. 1a as sites A and B, respectively), in agreement with previous findings[35]. For the remainder of this work, we focus on RIXS spectra recorded with the incident photon energies tuned to the UHB B (or A) peak, where an in-chain (or out-of-chain) O 1$s$ core electron is promoted into a neighbouring Cu 3$d$ orbital. This final state of the XAS process dictates the intermediate state of RIXS and is important in determining the scattering cross-section.

Figure 1c shows RIXS spectra measured with the incident photon energy tuned to the resonance of the A and B peak at the O $K$-edge in comparison with Cu $L_3$-edge data at $q = \pi/2a$. There are two energy regions with pronounced excitations: one below 1 eV and one above 1.5 eV, separated by a region of very weak spectral weight. The excitations at higher energies are dominated

by inter-orbital $dd$ and charge transfer (CT) excitations[4,7]; the $dd$ excitations are dominant at the Cu $L_3$-edge, whereas the CT excitations are dominant at the O $K$-edge.

Figure 1d zooms in on the low-energy excitation region, well below the $dd$ and CT excitations, which is our focus. O $K$ RIXS for photon energies tuned to B with different incident angles are compared to low-energy Cu $L_3$ RIXS data. Below 1 eV we see several excitations. In addition to the elastic line at zero energy transfer, we observe a weakly dispersing excitation at ~90 meV with varying cross-section for the different configurations. This behavior is typical of an optical phonon excitation and the energy scale agrees well with that of a Cu-O bond-stretching lattice vibration[15]. We, therefore, assign this feature to such a phonon. The line spectrum at $q = \pi/2a$ (see Fig. 1d, turquoise solid line) reveals a sharp structure coinciding with the very strong spinon excitations at the same $q$-point in Cu $L_3$-edge data (black line, note that the Cu $L_3$ spectrum is divided by a factor 10). In addition, the line spectrum taken close to the $\Gamma$-point (red line in Fig. 1d) is dominated by a broad structure, centered at ~0.5 eV and extending to about 1 eV in energy transfer. The energy of this structure is well separated from the $dd$ and CT excitations, suggesting that they are magnetic in origin. A possible path for creating magnetic excitations during RIXS at the O $K$-edge is sketched in Fig. 1e. This process will be discussed in more detail in the Discussion section.

To probe the dynamic character of the low-energy magnetic excitations visible in the O $K$-edge RIXS spectra, we have studied their momentum dependence for momentum transfer along the

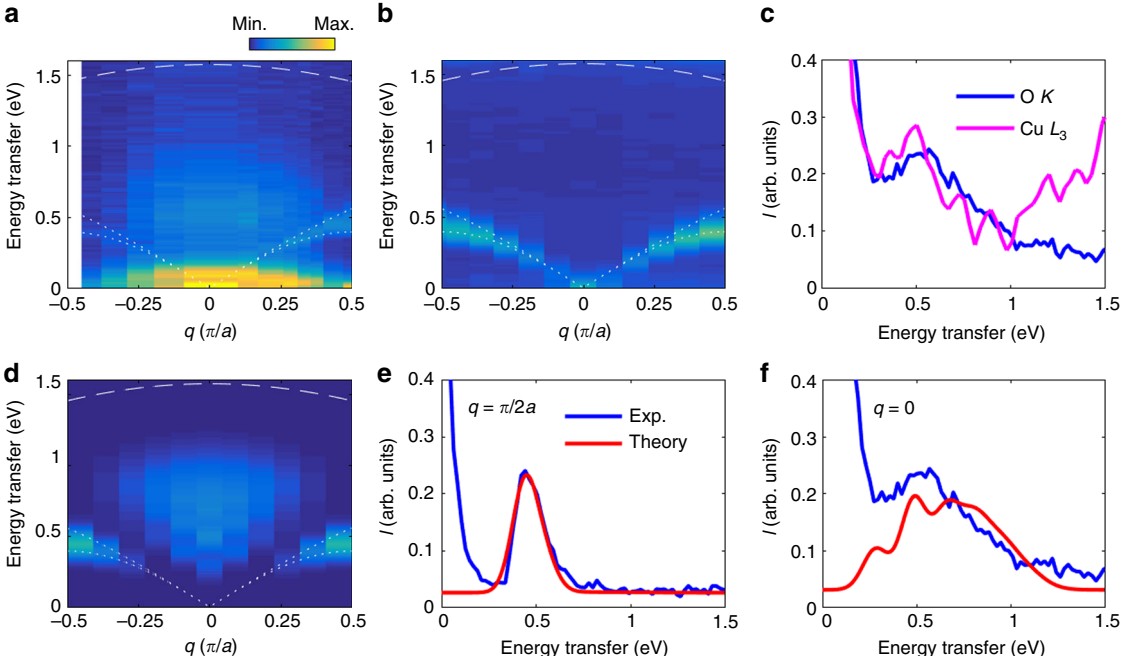

**Fig. 2** Comparison between the experimental and calculated RIXS spectra at the oxygen $K$-edge. **a**, **b** Experimental RIXS spectra as a function of momentum transfer and energy transfer measured at the oxygen $K$-edge with an incident photon energy of $\hbar\omega_{in} = 528.6$ eV (resonance B) (**a**) and at the copper $L_3$-edge from ref. [4] (**b**). **c** compares the O $K$-edge (blue) and Cu $L_3$-edge (magenta line) RIXS line cuts at $q = 0$. In the case of Cu $L_3$ data, there is a tailing contribution from higher energy $dd$ excitations, which extends down to low energy transfer (see Supplementary Note 3). **d** displays the calculated oxygen $K$-edge spectra for $\hbar\omega_{in} = 500$ meV. (This value optimizes the intensity of the four-spinon features, see Supplementary Note 5.) The excitation around 90 meV in the oxygen $K$-edge data is a phonon excitation not included in our model calculations. The modeled RIXS intensity was obtained from exactly diagonalizing a 22-site $t{-}J$ chain with periodic boundary conditions and the elastic line has been removed from the data for clarity. **e**, and **f** show line cuts of the RIXS spectra at $q = \pi/2a$ and $q = 0$, respectively. The experimental (theoretical) data are represented by the blue (red) solid lines. The intensity of the RIXS data are reported in arbitrary units (arb. units). The dotted and dashed white lines in **a**, **b**, and **d** indicate the boundaries of the two- and four-spinon continua, respectively (for more details on these boundaries see Supplementary Note 4)

chain direction, as shown in Fig. 2a. (O $K$-edge RIXS allows studying about 25% of the first Brillouin zone along [100] towards each side of $q = 0$, see Supplementary Figure 1.) The experimental geometry is described in Supplementary Note 1 and shown in Supplementary Figure 1. Additional comparisons of the data with the Cu $L_3$-edge are provided in Supplementary Notes 2 and 3.

In addition to the strong phonon excitation in O $K$-edge data, there are two distinct sets of continua in the magnetic region between 0.2 and 1.0 eV. One is dispersing towards zero energy for $q = 0$ and lies well within the boundaries of the two-spinon continuum (indicated by the white dotted lines). The second region is centered at $q = 0$ and 500 meV energy transfer and is clearly situated outside of the two-spinon continuum. The boundaries in Fig. 2 correspond to the two-spinon continuum expected for the 1D HAFM model obtained from purely kinematic constraints—see also Supplementary Note 4—assuming a superexchange value of $J = 250$ meV, as inferred from prior scattering experiments[4,8]. Comparison to Cu $L_3$ data displayed in Fig. 2b, where the two-spinon continuum dominates the spectrum, illustrates that O $K$-edge and Cu $L_3$-edge RIXS have quite different responses in terms of the magnetic excitations. However, the line cuts of O $K$-edge and Cu $L_3$-edge RIXS spectra in Fig. 2c show that there is also a finite weight in Cu $L_3$-edge RIXS spectra at $q = 0$. Note that the O $K$-edge data reveal much stronger polarization dependence due to difference in connectivity of the in-chain and out-of-chain O $2p$ orbitals (see Supplementary Note 1).

The fact that the $\Gamma$-point excitations appear outside the boundaries of the two-spinon continuum, and well below the

energy transfers where $dd$ and CT excitations occur suggests that they are multi-spinon in nature. This interpretation is further supported by the fact that they lie completely within the boundaries expected for the four-spinon continuum (indicated by dashed lines, as obtained from pure kinematic arguments—see Supplementary Note 4)[24].

**Theoretical results.** The phase space considerations given above identify the $\Gamma$-point excitations belonging to multi-spinon excitations involving at least four or more spinons. However, to understand the spectral weight of these excitations, it is necessary to compute the RIXS intensity within the Kramers–Heisenberg formalism, due to the prominent role played by the core-hole lifetime at the O $K$-edge. To this end, we performed small cluster exact diagonalization calculations to further elucidate the nature of these excitations. Since we are interested in the energy region well below the $dd$ and CT excitations, we used the $t{-}J$ model, where these multi-orbital processes have been integrated out[36]. In this case, we adopt values of the hopping integral $t = 300$ meV and superexchange interaction $J = 250$ meV, which are consistent with existing literature (see Methods). The computed spectra (with elastic peak removed) are compared against the experimental data in Fig. 2d. Line cuts of the data superimposed over the calculations are shown in Fig. 2e ($q \approx \pi/2a$) and Fig. 2f ($q = 0$).

The overall agreement between the calculated magnetic response and the experimental data is excellent: our model captures both the dispersing magnetic excitations and the broad

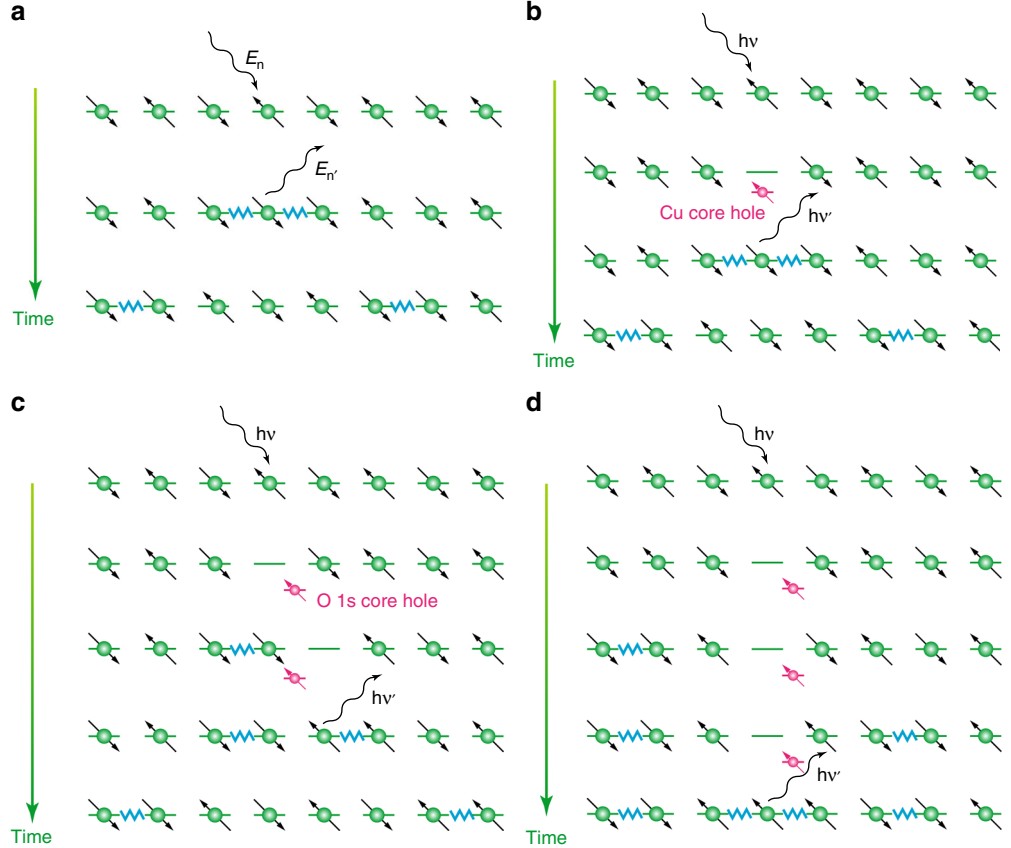

**Fig. 3** An illustration of various spin excitations through elementary spin-flips. **a** The $\Delta S_{tot}=1$ direct spin-flip process that can occur in an inelastic neutron scattering experiment, which primarily decays into two-spinon excitations that are visualized as domain walls in the antiferromagnetic (AFM) background[5]. **b** The same $\Delta S_{tot}=1$ spin-flip process in RIXS, which is accessible in materials with strong spin–orbit coupling in the core level[32]. **c** The indirect double spin-flip process at the oxygen $K$-edge, which occurs via the multi-orbital hopping processes sketched in Fig. 1e. This process generates a nearest-neighbour double spin flip, which predominantly decays into a two-spinon excitation[22,38]. **d** A second-order process at the oxygen $K$-edge that produces four-spinon excitations. Here, the absence of the spin in the AFM chain allows double spin-flips to occur on the sites adjacent to the missing spin. These double spin-flips generate spinon excitations away from the site where the core hole is created. The subsequent decay of the core hole then produces two additional spinons in its vicinity. This process requires a long-lived core-hole to allow for sufficient time to generate the two double spin-flips before the core-hole decay occurs

continuum centered at the $\Gamma$-point. Even the quantitative agreement is very good. Note that our model does not contain the lattice degrees of freedom, and so it does not reproduce the peak associated with the lattice excitations. (If the spin–lattice coupling is weak, we expect that the inclusion of the lattice vibrations would superimpose a phonon excitation on the RIXS spectrum). In this case, the level of agreement between the model and the data in the magnetic region indicates that any spin–lattice coupling is small and that the final states of the O $K$-edge RIXS process can be well described solely by excitations of the half-filled $t-J$ model, whose final states are the same as those in the Heisenberg model. This observation justifies our neglecting of the lattice excitations and allows us to identify the upward dispersing branch as two- and four-spinon excitations, commonly associated with DSF for $\Delta S_{tot} = 1$, while the continuum of excitations centered at $q = 0$ corresponds to four-spinon excitations that require a more complex correlation function. This assignment is further supported by the dependence of these excitations on the core-hole lifetime, which will be discussed shortly.

**Discussion**

How can we understand the magnetic excitations in RIXS captured by the $t-J$ model, and why do we see magnetic excitations

that are absent in INS? In Fig. 3 we illustrate schematically the magnetic excitation mechanisms in a spin chain with the different scattering techniques: INS (Fig. 3a), Cu $L_{2,3}$-edge (Fig. 3b), and O $K$-edge RIXS (Fig. 3c, d). The ground state of the HAFM is a SU (2)-symmetric singlet with $S_{tot} = 0$. INS measurements connect this ground state to the triplet manifold with $S_{tot} = 1$ at low temperatures, such that $\Delta S_{tot} = 1$. In a simplified picture (shown here for $\Delta S_z = \pm 1$), such excitation creates two domain walls in the spin chain (Fig. 3a), which decay predominantly into two-spinons carrying parallel spins of ½ (due to conservation of angular momentum). Exact calculations[24] show that these excitations also have overlap with four-spinon excitations, but the majority of the four-spinon weight remains within the boundaries of the two-spinon continuum due to kinematic constraints in the matrix elements.

Unlike INS (and RIXS at the Cu $L_{2,3}$-edges), excitations with $\Delta S_{tot} = 1$ are generally forbidden for $K$-edge RIXS. (This statement holds only for materials with small spin–orbit coupling in the valence band; single flips are allowed in O $K$-edge RIXS on iridates, see ref. [37].) Instead, $\Delta S_{tot} = 0$ processes like the one sketched in Fig. 1e must be used to create magnetic excitations. Here, the incident photon creates a Cu $3d^{10}$ UHB excitation in the intermediate state, resulting in a Cu site with an additional

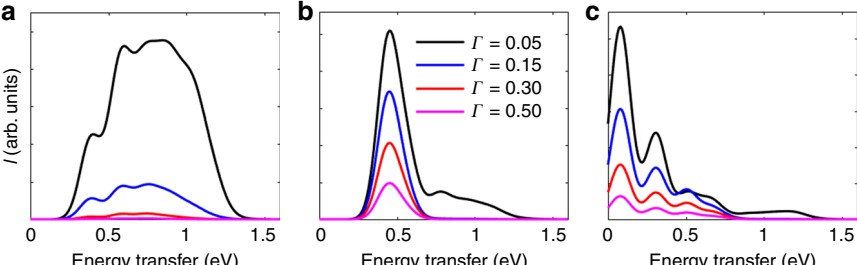

**Fig. 4** The effect of the core-hole lifetime on the RIXS spectra. The variation in the computed RIXS intensity at momentum transfer **a** $q = 0$, **b** $q = \pi/2a$, and **c** $q = \pi/a$. As the core-hole lifetime is decreased (increasing $\Gamma$), the four-spinon excitations at $q = 0$ disappear rapidly, while the two-spinon contributions to the spectra at $q = \pi/2a$ and $q = \pi/a$ are more robust. The intensity scale is reported in arbitrary units (arb. units)

electron in direct vicinity to an O $1s$ core hole (represented in the sketch as blue arrow (spin down) on the left-hand Cu side). The 180° Cu-O-Cu bonding angle in $Sr_2CuO_3$ enables efficient double inter-site hopping of $3d$ electrons between two adjacent Cu sites via the bridging in-chain oxygen site (B in Fig. 1a), transferring the Cu $3d^{10}$ to the neighbouring Cu site (right-hand Cu side in Fig. 1e). Since this Cu atom is also hybridized with the oxygen where the core hole is localized, the electron represented by the blue arrow can then decay and fill the core level (see Fig. 1e), leaving the system with a net double inter-site spin flip. This process, sketched in Fig. 3c, is analogous to an indirect double spin-flip process predicted for Cu $K$ RIXS[38,39] giving rise to a double domain wall that decays predominantly into two spinons carrying antiparallel spins (due to momentum conservation)[22]. The cross-section can be related to a dynamic exchange correlation function, whose spectral weight is similar to that of DSF near the zone center[38]. This excitation pathway explains the presence of the sharper dispersing magnetic excitations in O $K$-edge RIXS spectra. (Note that in two dimensions a $\Delta S_{tot} = 0$ excitation can only create bi-magnon excitations, as each magnon carries a spin of 1). To visualize the scattering process responsible for the creation of four-spinon excitations around the $\Gamma$-point the lifetime of the intermediate state plays a critical role.

As we mentioned previously, in Cu $L_{2,3}$-edge RIXS $\Delta S_{tot} = 1$ spin-flip excitations that are similar to INS are allowed[32]. This is possible, since for a Cu $2p$ core-hole the spin–orbit coupling is strong and therefore the change of spin momentum can be compensated by the change of angular momentum. In contrast to INS, however, RIXS involves a doublon in the intermediate state, which decays on a timescale set by the core-hole lifetime (~several femtosecond)[40]. During this time, the additional charge in the intermediate state can interact with the system, creating excitations that are inaccessible through either a single or double spin-flip process. For the O $1s$ core hole there is no appreciable angular momentum available; therefore, the spin momentum must be conserved and only $\Delta S_{tot} = 0$ excitations are possible (as described above) (Fig. 3c). In a 1D system, the result of this $\Delta S_{tot} = 0$ excitation looks very similar to the result of a single spin flip $\Delta S_{tot} = 1$ in that both excitations lead to the creation of two domain walls, but at the O $K$-edge they are separated by at least one atomic site and have opposite spins. The lifetime of O $1s$ core-hole states is somewhat longer than the lifetime of Cu $2p$ core-hole states, however. During this time, the doublon in the $3d$ band can also generate double spin-flips on the surrounding sites, as sketched in (Fig. 3d), creating two additional double spin-flips separated by larger lattice distances. The subsequent decay of the core hole results in the creation of two additional domain walls, and a total of four spinons in the final state. This scattering channel is the direct result of fluctuations that take place in the intermediate state. Its intensity, therefore, depends on the lifetime of the core-hole, as a longer-lived doublon will have sufficient

time to generate the longer-range double spin-flips, separated by large lattice distances. This excitation channel is expected to be weak in Cu $L_3$ RIXS, whose core-hole is short lived, and completely absent in INS.

We performed calculations for the dependence of these excitations on the lifetime of the intermediate state to test our interpretation. The results are presented in Fig. 4. We observe that upon decreasing the core-hole lifetime (increasing $\Gamma$), the intensity of magnetic excitations in O $K$-edge RIXS decreases. Moreover, the spectral weight of the four-spinon excitations moves towards smaller energy transfers (see Fig. 4a). The decrease in intensity is much slower for excitations belonging to the two-spinon continuum than for the four-spinon excitations. Whereas the two-spinon excitations are still quite pronounced for $\Gamma = 500\,meV$ (Fig. 4b, c), the four-spinon excitations are suppressed below $\Gamma = 300\,meV$ (Fig. 4a), which is comparable to the super-exchange interaction $J$. The suppression of the four-spinon weight at $q = 0$ proves that the core-hole lifetime sets the time scale for the intermediate state to generate these excitations. As its lifetime is quenched below $J$ (~1.3 fs), there is not enough time for additional double spin-flips to occur in the chain during the frustrating presence of the doublon. The dynamics of this intermediate state plays an important role for the discovered excitation channel for magnetic excitations and produces additional magnetic correlation functions—beyond a single or double spin flip.

We have demonstrated that RIXS grants access to complementary correlation functions for magnetic scattering compared to INS, which arises from the lifetime and dynamics of the intermediate state. Importantly, this scattering channel is unique to RIXS and provides access to non-local spin correlation functions beyond two-site correlation functions probed by traditional scattering techniques. O $K$-edge RIXS has long core-hole lifetimes and is therefore ideal for examining excitations that cannot be detected by INS scattering, as long lifetimes of the intermediate state allow spin and charge fluctuations to take place. We have exploited this fact to observe *directly* four-spinon excitations of a pure 1D HAFM, located outside the boundaries of the two-spinon continuum. This technique opens another avenue to explore quantum magnetism and quasi-particle fractionalization, which has broad applications in the field of quantum magnetism. Time-resolved studies at the upcoming XFEL sources, for example, European XFEL and Swiss FEL, will hopefully facilitate studying such dynamics at the femtosecond timescale.

## Methods

**Experiment**. We applied the technique of high-resolution RIXS with the incident photon energy tuned to the O $1s$ core → $2p$ UHB resonance (around 528 eV). Single-crystal samples of $Sr_2CuO_3$ were grown by the floating-zone method and freshly cleaved before the RIXS experiment. During the experiment the surface normal to the sample, [010], and the propagation direction of the chains, [100], were oriented parallel to the scattering plane. The scattering plane was horizontal.

# ARTICLE

The sample was cooled with a helium-flow cryostat to 14K during the measurements. The experiments were performed at the ADRESS beamline (BL) of the Swiss Light Source at the Paul Scherrer Institut[41,42]. Incident photons were linearly polarized either in the scattering plane ($\pi$-polarization), which was the case for most of the data, or perpendicular to the scattering plane ($\sigma$-polarization). The XAS data were measured in total fluorescence yield. The BL energy resolution was set to 70 meV or better, with the BL exit slit open to 30 μm. (The BL energy resolution for the Cu $L_3$ data[4] was 100 meV or better, with the BL exit slit open to 10 μm.) The SAXES RIXS spectrometer was located at a fixed scattering angle of $\Psi = 130° \pm 1°$, whereas the incidence angle on the sample varied between $10° \pm 1°$ and $110° \pm 1°$ grazing (see Supplementary Figure 1). The angular horizontal acceptance of the spectrometer was approximately 5 mrad[40]. The total experimental energy resolution was 80 meV and the simultaneously recorded energy window was 22.2 eV (the total experimental resolution for the Cu $L_3$ data[4] was 140 meV and the simultaneously recorded energy window was 59.2 eV).

**Cluster calculations**. The RIXS intensity $I(q, \Omega)$ was evaluated using the Kramers–Heisenberg formalism where ($\hbar = 1$)

$$I(q, \Omega) = \sum_f \left| \sum_{n, R_m} e^{-iqR_m} \frac{\langle f|D_m^\dagger|n \rangle \langle n|D_m|i \rangle}{E_i + \omega_{in} - E_n + i\Gamma} \right|^2 \delta\left(E_f - E_i + \Omega\right), \quad (1)$$

Here, $q = \mathbf{e_x} \cdot (\mathbf{k_{out}} - \mathbf{k_{in}})$ is the momentum transfer along the $x$-axis and $\Omega = \omega_{out} - \omega_{in}$ is the energy loss, $D$ is the dipole operator, and $|i\rangle$, $|n\rangle$, and $|f\rangle$ are the initial, intermediate, and final states of the RIXS process with energies $E_i$, $E_m$, and $E_f$, respectively, $R_m = am$ is the position of the $m$th Cu atom, $a$ is the Cu–Cu distance, and $\Gamma$ is the core-hole lifetime. We compute the eigenstates by diagonalizing $t-J$ Hamiltonian defined on a 22 site cluster. The use of this low-energy effective model is justified since all of the $dd$ and CT excitations appear well above 1 eV in energy loss (see Fig. 1c). Moreover, recent DMRG calculations have explicitly shown that the magnetic excitations probed by Cu $L$-edge RIXS obtained from a four orbital $pd$ model for $Sr_2CuO_3$ can be accurately reproduced using an effective $t-J$ Hamiltonian[36] up to an overall rescaling of the intensity. This result gives us confidence that the downfolded $t-J$ Hamiltonian can capture the magnetic excitations of $Sr_2CuO_3$.

The dipole operator in the effective model is given by

$$D_m = \sum_\sigma \left(d_{m,\sigma} - d_{m+1,\sigma}\right) s_{m,\sigma}^\dagger, \quad (2)$$

where $d_{m,\sigma}$ annihilates a spin $\sigma$ hole on Cu site $m$ and $s_{m,\sigma}^\dagger$ creates a hole in the oxygen $1s$ orbital on the site between the $m$ and $m+1$ Cu sites. Here, the relative phases reflect the phases of the original Cu-O overlap integrals. The model parameters are $t = 300$ meV and $J = 250$ meV, which is appropriate for $Sr_2CuO_3$[4,7], and $\Gamma = 150$ meV for the oxygen $K$-edge[15,19].

**Code availability**. The source code for the RIXS calculations is available from the authors upon reasonable requests. Requests for the code should be directed to S.J.

## Data availability

The data that support the findings of this study are available from the authors upon reasonable requests. Requests for experimental data should be directed to J.S. and T.S.

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

## Acknowledgements

We thank C. Batista and K. Wohlfeld for useful discussions. The experiments were performed at the ADRESS BL of the Swiss Light Source at the Paul Scherrer Institut. We acknowledge support from the Swiss National Science Foundation and its NCCR MaNEP. CPU time was provided in part by resources supported by the University of Tennessee and Oak Ridge National Laboratory Joint Institute for Computational Sciences (http://www.jics.utk.edu).

## Author contributions

J.S. and T.S. designed the experiment. J.S., T.S., and K.J.Z. performed the experiment with the assistance of V.N.S. M.M., H.M.R., and L.P. contributed to the discussion of data. J.S. performed data analysis in discussion with T.S., U.K., and S. J. performed theory calculations. S.S. and A.R. have prepared and characterized the crystal samples. T.S. and S.J. were responsible for project management. J.S. wrote the paper together with U.K., S.J., and T.S. with input from all other authors.

## Additional information

**Competing interests:** The authors declare no competing interests.

