## [Peer Review File · Nature Communications]

Reviewers' comments:

Reviewer #1 (Remarks to the Author):

The authors have addressed my comments satisfactorily.

Reviewer #2 (Remarks to the Author):

This work presents an experimental study of magnetic excitations in the quasi-1D spin-chain cuprate Sr₂CuO₃ using oxygen K-edge resonant inelastic x-ray scattering. The authors observed that magnetic excitations exist in two non-overlapping regions of phase space, one predominantly corresponding to the two-spinon continuum and one to the four-spinon continuum. While the two-spinon continuum in this compound has been previously reported by inelastic neutron scattering and also by RIXS, the authors claim that four-spinon continuum in this compound has not been yet reported, and I agree with the authors that the observation of four-spinon continuum and especially its separation from the two-spinon continuum is usually not a trivial task. Previously, four-spinon continua have been reported outside of the two-spinon continuum in Yb₂Pb₂Pb and the 1D ferromagnet LiCuVO₄. In the current manuscript, the authors show that the RIXS intensity at Gamma point contains no contribution from the two-spinon continuum and, therefore, must come from the multi-spinon continuum.

Overall, I find this work interesting and worth publishing in Nature Communications. However, I still find that the manuscript is not completely clear even after the revision of the manuscript, which probably led to the improvement of the text with respect to the original one. I think that the authors put more efforts in the answering to the previous reviewers than to the revising the text of the manuscript. I find that several important aspects of the work can be more easily understood from the author's reply letter than from the main text of the manuscript.

I found a few sentences throughout the text simply confusing. For example, The authors say that the "overall agreement between the calculations and the experimental data is excellent" in Fig. 2 (d) and (f). However, the excellent agreement is only observed above a particular frequency range (as it should be) but this should be written explicitly. Also, authors stressed that the phonon excitation is not

included in the theory however they provided no arguments about how the inclusion of the phonon in the theory would modify the results.

Also, the authors did not justify their choice of the model. Is it known that all interactions along the chain are the same? Is it safe to neglect farther neighbor interactions in modeling this compound? Some clarifications should be done in the text.

Reviewer #3 (Remarks to the Author):

Review of NCOMMS-18-17877-T "Direct observation of multi-spinon excitations outside of the two-spinon continuum in the antiferromagnetic spin-1/2 chain cuprate Sr₂CuO₃" by Schlappa et al.

Using RIXS, this paper reports an observation of magnetic excitations in the 1D S=1/2 Heisenberg antiferromagnetic chain (HAFC) material Sr₂CuO₃, showing clear evidence for excitations outside the regime where fractional excitations in the S=1/2 HAFC have been typically detected by inelastic neutron scattering. The regime where these excitations have been measured is little explored in the literature, and the technique for measuring them is innovative, so overall my feeling is that this contribution has enough new material to warrant publication in Nature Communications. As explained below I have some reservations about the way the paper is written

that I believe should be addressed by the authors before publication.

Before proceeding, I note that I have received a copy of the current version of the manuscript and supplementary material, as well as the authors' responses to three previous referees. Those responses include excerpts from the referees' reports but I have not necessarily seen those reports in their entirety.

Now to the scientific issues. Perhaps it is not widely appreciated just how well the $S=1/2$ HAFC is understood. A deep knowledge of the fractional excitations in this system has been built up over decades, both through theoretical efforts including Bethe Ansatz and field theory, as well as experiments on many systems, notably KCuF_3 starting with time of flight neutron measurements in the early 1990's. As written the paper overemphasizes the importance of "4-spinon" excitations, and also contains misleading statements about what is measured by inelastic neutron scattering.

Inelastic neutron scattering in the pure $S=1/2$ HAFC at $T=0$ measures matrix elements connecting the ground state (known to be a singlet) with the first manifold of excited states, which forms a triplet continuum. There is an equal contribution to the scattering from each of three pieces corresponding to $\Delta S = -1$, $\Delta S = 0$, and $\Delta S = +1$.

As is explained in reference 22 (Caux and Hagemans, 2006) of the manuscript (and alluded to in other papers) the eigenstates of the Hamiltonian can be written as superpositions of basis states consisting of contributions with even numbers of spinons, i.e. 2 spinon, 4 spinon, 6 spinon ... etc. It turns out that the boundaries of the portion of energy-momentum space spanned by the first manifold of excited states, with total spin $S=1$, are identical to the boundaries that would be inferred from the kinematical constraints imposed by simply considering 2 spinons. The eigenstates themselves contain contributions from 2, 4, 6, even numbers of spinons. In this first manifold approximately 73% of the total intensity comes from the part of the squared matrix elements related to the 2 spinon contribution to the expansion(s), and most of the rest comes from the 4 spinon contribution, with some additional contributions from higher order contributions. All of these are sampled by neutron scattering. There is nothing magical about 2 spinon contributions vs. 4 spinon etc.

The manifold of states with a total spin $S=2$ contributes little or nothing to the inelastic neutron scattering response at $T=0$ unless the manifolds are mixed somehow. However there will be contributions at $T > 0$ from transitions between manifolds with $S=1$ and $S=2$. The same holds true for other manifolds.

Historically much was understood about the Bethe Ansatz solution for the eigenstates but the calculation of matrix elements connecting them was extremely complex and too difficult to carry out. This is where there has been significant progress in more recent years, notably by Caux and colleagues.

As currently written this paper puts a great deal of emphasis on the apparent novelty of the observation of "4 spinon" states. The observations presented in the paper are very nice, but in reality the 4 spinon terms and the associated kinematic boundaries are quite well understood. Where the new measurements reported here can lend insight into the physics is by providing a way to compare experiment to a set of matrix elements complementary to those measured by neutron scattering, albeit with a more complicated cross-section. In principle this can provide an additional test for the modern methods of calculating response function and thus is a challenge to theory.

The paper contains misleading statements about inelastic neutron scattering, including claims that the " $\Delta S = 0$ channel is not often used to probe magnetic excitations". As seen from the discussion above, this claim is demonstrably false when it comes to the HAFC, or any other example involving singlet to triplet transitions. Perhaps the authors are thinking of typical conventional spin waves in insulators, which are generally transverse and involve $\Delta S = -1, 1$, whereas 2 magnon scattering is dominated by $\Delta S = 0$ matrix elements. The direct observation of 2 magnon states by light scattering is in some ways analogous to the RIXS results reported here.

To summarize, I endorse publishing this paper in Nature Communications, but ask the authors to fix some of the statements about neutron scattering and maybe tone down the claims about the novelty of the observation of "4 spinon" states.

Response to Reviewer #1

The Reviewer wrote: The authors have addressed my comments satisfactorily.

Our reply: We thank the reviewer for her/his time in examining our work a second time and for their positive recommendation.

Response to Reviewer #2

The Reviewer wrote: This work presents an experimental study of magnetic excitations in the quasi-1D spin-chain cuprate Sr₂CuO₃ using oxygen K-edge resonant inelastic x-ray scattering. The authors observed that magnetic excitations exist in two non-overlapping regions of phase space, one predominantly corresponding to the two-spinon continuum and one to the four-spinon continuum. While the two-spinon continuum in this compound has been previously reported by inelastic neutron scattering and also by RIXS, the authors claim that four-spinon continuum in this compound has not been yet reported, and I agree with the authors that the observation of four-spinon continuum and especially its separation from the two-spinon continuum is usually not a trivial task. Previously, four-spinon continua have been reported outside of the two-spinon continuum in Yb₂Pb₂Pb and the 1D ferromagnet LiCuVO₄.

In the current manuscript, the authors show that the RIXS intensity at Gamma point contains no contribution from the two-spinon continuum and, therefore, must come from the multi-spinon continuum. Overall, I find this work interesting and worth publishing in Nature Communications.

Our reply: We thank the reviewer for their time in reviewing our manuscript, and for their positive evaluation attesting that our work is of sufficient importance for Nature Communications.

The reviewer wrote: However, I still find that the manuscript is not completely clear even after the revision of the manuscript, which probably led to the improvement of the text with respect to the original one. I think that the authors put more efforts in the answering to the previous reviewers than to the revising the text of the manuscript. I find that several important aspects of the work can be more easily understood from the author's reply letter than from the main text of the manuscript.

Our reply: We thank the reviewer for pointing out this lack of clarity. We have gone through the manuscript again with an eye to compare it to our reply to the original round of referee reports. We introduce now several of the explanations from the answers to the previous referees into an updated version of our manuscript where we felt it was appropriate or where it would improve clarity. We are confident that this process has improved the presentation of our work.

The reviewer wrote: I found a few sentences throughout the text simply confusing. For example, The authors say that the "overall agreement between the calculations and the experimental data is excellent" in Fig. 2 (d) and (f). However, the excellent agreement is only observed above a particular frequency range (as it should be) but this should be written explicitly.

Our reply: We thank the reviewer for pointing out this issue. Indeed, our results agree in terms of the magnetic part of the RIXS spectra and our model does not include the phonon

degrees of freedom and hence does not capture the near-elastic features. In the new version of the manuscript we have qualified our statement and added discussion about neglecting phonons in our model (see below).

The reviewer wrote: Also, authors stressed that the phonon excitation is not included in the theory however they provided no arguments about how the inclusion of the phonon in the theory would modify the results.

Our reply: We agree with the referee that this could be more clearly stated. Some of the current authors have studied phonons previously using RIXS and so this point was clear to us but not to a general reader. The phonon excitation observed here is an optical phonon with an energy ~ 90 meV. In the RIXS spectra, such excitations appear as nearly dispersionless features at the phonon energy and hence are well separated from and easily distinguishable from the spin excitations. Therefore, the magnetic excitations will be relatively unaffected by the inclusion of phonons in the model. In principle the inclusion of significant spin-phonon interactions might modify the results. However, as a pure t-J model proved sufficient to explain both the INS and RIXS data, it was unnecessary to assume such spin-phonon interactions were present. For clarity, we added a statement in the revised manuscript stating that the spin degree of freedom in this spin chain system is sufficient to explain the observed magnetic excitations in the RIXS data and we have included references to systems where phonons have been observed before.

The reviewer wrote: Also, the authors did not justify their choice of the model. Is it known that all interactions along the chain are the same? Is it safe to neglect farther neighbor interactions in modeling this compound? Some clarifications should be done in the text.

Our reply: There are two issues here that should be addressed. The first is our choice in the t-J model parameters, and the second is the use of the t-J model itself over the full multi-orbital model. In response to the first issue, we note that the parameters for the t-J model of Sr_2CuO_3 are now fairly well established in the literature by prior experimental studies. For example, Walters, *et. al.*, [Nature Physics **5**, 867 (2009)] and Schlappa, *et. al.*, [Nature **485**, 82 (2012)] described inelastic neutron scattering and Cu L-edge RIXS data, respectively, with just isotropic nearest neighbour spin interactions. Similarly, fits to the INS data indicate a negligible next nearest neighbour exchange interaction is present in the system. The only relevant longer-range exchange interaction is a weak interchain coupling, which is responsible for the long-range antiferromagnetic ordering below $T = 5$ K; however, above this temperature (e.g. for our measurements at 14 K) this interaction can be neglected.

In response to the second point, we note that a recently published DMRG study [A. Nocera *et al.*, Scientific Reports **8**, 11080 (2018)] by some of the authors showed that the downfolded t-J model is adequate for describing the magnetic excitations observed in RIXS. Specifically, they compared the magnetic RIXS response at the Cu L-edge computed using a full multi-orbital Cu-O model for Sr_2CuO_3 directly with the RIXS response computed with the t-J model. They found that the two models give identical results apart from an overall scaling factor, which is related to the degree of covalency in the model. This result, in combination with the level of agreement we have obtained here, gives us great confidence that the t-J model with isotropic nearest neighbour interactions only captures the magnetic excitations at the O K-edge. The DMRG study had not been published or posted on the arXiv when we first submitted this paper so we did not cite it. However, in the current version we have added

discussion along these lines and a citation to that work to justify our approach in the current work.

Response to Reviewer #3

The Reviewer wrote: Review of NCOMMS-18-17877-T "Direct observation of multi-spinon excitations outside of the two-spinon continuum in the antiferromagnetic spin-1/2 chain cuprate Sr₂CuO₃" by Schlappa et al. Using RIXS, this paper reports an observation of magnetic excitations in the 1D S=1/2 Heisenberg antiferromagnetic chain (HAFC) material Sr₂CuO₃, showing clear evidence for excitations outside the regime where fractional excitations in the S=1/2 HAFC have been typically detected by inelastic neutron scattering. The regime where these excitations have been measured is little explored in the literature, and the technique for measuring them is innovative, so overall my feeling is that this contribution has enough new material to warrant publication in Nature Communications. As explained below I have some reservations about the way the paper is written that I believe should be addressed by the authors before publication.

Our reply: We thank the reviewer for their time in reviewing our work and for their obvious interest in the material. We are also encouraged by their insightful comments and overall positive recommendations for publication.

The reviewer wrote: Before proceeding, I note that I have received a copy of the current version of the manuscript and supplementary material, as well as the authors' responses to three previous referees. Those responses include excerpts from the referees' reports but I have not necessarily seen those reports in their entirety.

Our reply: It is our internal policy to quote all referee reports in their entirety when responding to referee comments to ensure that the proper context is provided. Therefore, our response to referees had all the excerpts from the referee unless some material has been censored by the editor.

The reviewer wrote: Now to the scientific issues. Perhaps it is not widely appreciated just how well the S=1/2 HAFC is understood. A deep knowledge of the fractional excitations in this system has been built up over decades, both through theoretical efforts including Bethe Ansatz and field theory, as well as experiments on many systems, notably KCuF₃ starting with time of flight neutron measurements in the early 1990's. As written the paper overemphasizes the importance of "4-spinon" excitations, and also contains misleading statements about what is measured by inelastic neutron scattering.

Our reply: We completely agree with the reviewer that S=1/2 HAFC has been extensively studied in the literature and is very well understood. This fact was one of the reasons why we found the four-spinon excitations observed here so interesting – they constituted newly observed excitations for this well understood system. We explicitly cite now in the updated version of our manuscript three of the first and very important INS experiments on such S=1/2 HAFC performed on KCuF₃ [D. A. Tennant et al., Physical Review B **52**, 13368 (1995), D. A. Tennant et al. Phys. Rev. Lett. **70**, 4003 (1993), and B. Lake et al., Nature Materials **4**, 329 (2005)].

The reviewer wrote: Inelastic neutron scattering in the pure S=1/2 HAFC at T=0 measures matrix elements connecting the ground state (known to be a singlet) with the first manifold of excited states, which forms a triplet continuum. There is an equal contribution to the

scattering from each of three pieces corresponding to $\Delta S = -1$, $\Delta S = 0$, and $\Delta S = +1$.

Our reply: We believe we are in agreement with the referee on these points, with a point of clarification. The singlet and triplet continuum for pure $S=1/2$ HAFC have net spin $S_{\text{tot}} = 0$ and $S_{\text{tot}} = 1$, respectively. Therefore, INS between these two manifolds must involve $\Delta S_{\text{tot}} = 1$. We think that possibly the referee meant $\Delta S_z = 0, \pm 1$, which are the states with equal contributions to the triplet manifold for INS scattering. On the other hand, in RIXS the cross-section at the oxygen K-edge of Sr_2CuO_3 forbids transitions to the triplet manifold with $S_{\text{tot}} = 1$. This is because (in the absence of strong spin-orbit coupling), RIXS cannot introduce a net spin-flip. Thus, the excitations we are observing are restricted to the singlet $S_{\text{tot}} = 0$ channel. When we wrote our paper, we had in mind the spin-flip channel for INS, where the neutron undergoes a $\Delta S_z = \pm 1$ transition and therefore so does the chain. We agree that this perspective could be misleading to the general reader and have revised this portion of the text to clarify matters. In particular, now we stress out clearly in the relevant passages of the manuscript that O K edge RIXS is probing singlet excited states of 1D HAFM. We have also replaced S by S_{tot} , to make it more clear that it refers to the total spin.

The reviewer wrote: As is explained in reference 22 (Caux and Hagemans, 2006) of the manuscript (and alluded to in other papers) the eigenstates of the Hamiltonian can be written as superpositions of basis states consisting of contributions with even numbers of spinons, i.e. 2 spinon, 4 spinon, 6 spinon ... etc. It turns out that the boundaries of the portion of energy-momentum space spanned by the first manifold of excited states, with total spin $S=1$, are identical to the boundaries that would be inferred from the kinematical constraints imposed by simply considering 2 spinons. The eigenstates themselves contain contributions from 2, 4, 6, even numbers of spinons. In this first manifold approximately 73% of the total intensity comes from the part of the squared matrix elements related to the 2 spinon contribution to the expansion(s), and most of the rest comes from the 4 spinon contribution, with some additional contributions from higher order contributions. All of these are sampled by neutron scattering. There is nothing magical about 2 spinon contributions vs. 4 spinon etc. The manifold of states with a total spin $S=2$ contributes little or nothing to the inelastic neutron scattering response at $T=0$ unless the manifolds are mixed somehow. However there will be contributions at $T > 0$ from transitions between manifolds with $S=1$ and $S=2$. The same holds true for other manifolds.

Our reply: We completely agree with the referee and thank them for their comments. Based on these points, we have revisited the text to describe the contributions to the dynamical structure factor measured by neutron scattering more accurately. The $S = 2$ manifold is also irrelevant for the RIXS cross section at low temperatures because the total spin is conserved at the oxygen K-edge. This means that the RIXS process can only link the $S = 2$ manifold back to itself, which will only be relevant at finite temperatures. Our experiment is conducted at low-temperature and the data is well captured by our zero temperature calculations. Therefore, we believe that the high spin manifolds can be neglected here.

The reviewer wrote: Historically much was understood about the Bethe Ansatz solution for the eigenstates but the calculation of matrix elements connecting them was extremely complex and too difficult to carry out. This is where there has been significant progress in more recent years, notably by Caux and colleagues. As currently written this paper puts a great deal of emphasis on the apparent novelty of the observation of “4 spinon” states. The

observations presented in the paper are very nice, but in reality the 4 spinon terms and the associated kinematic boundaries are quite well understood.

Our reply: We again agree with the referee that the 4 spinon contributions to the dynamical spin structure factor are well understood. Indeed, as Caux and his colleagues have shown, there is even a small amount of four spinon weight in $S(q, \omega)$ that is located between the upper boundary of the two spinon continuum and the upper boundary of the four spinon continuum. Our measurement, however, detects spectral weight that is inconsistent with that expected for $S(q, \omega)$, which we have explained in terms of differences in the RIXS scattering amplitude in comparison to INS. We believe that this observation is in itself important, as we will explain below.

The reviewer wrote: Where the new measurements reported here can lend insight into the physics is by providing a way to compare experiment to a set of matrix elements complementary to those measured by neutron scattering, albeit with a more complicated cross-section. In principle this can provide an additional test for the modern methods of calculating response function and thus is a challenge to theory.

Our reply: We again agree with the reviewer that our work provides access to a set of matrix elements complementary to INS, and that our oxygen K-edge RIXS data of Sr_2CuO_3 can provide an additional test for the theory. But we also believe that detecting four-spinon excitations beyond those expected in INS [and encoded in $S(q, \omega)$] is also important. For example, as the reviewer is no doubt aware of, there is a large effort currently underway searching for materials realizations of the Kitaev model and its associated quantum spin liquid ground state. In most candidate materials, however, long range magnetic order is often stabilized by residual Heisenberg interactions. In these cases, the community has begun studying the high-temperature excited states using INS to search for signatures of the Kitaev physics. Our results show that RIXS has access to complementary spinon excitations, also appearing at high-energy, and we believe that the technique could be used to facilitate the search for Kitaev physics.

The reviewer wrote: The paper contains misleading statements about inelastic neutron scattering, including claims that the “ $\Delta S = 0$ channel is not often used to probe magnetic excitations”. As seen from the discussion above, this claim is demonstrably false when it comes to the HAFC, or any other example involving singlet to triplet transitions. Perhaps the authors are thinking of typical conventional spin waves in insulators, which are generally transverse and involve $\Delta S = -1, 1$, whereas 2 magnon scattering is dominated by $\Delta S = 0$ matrix elements. The direct observation of 2 magnon states by light scattering is in some ways analogous to the RIXS results reported here.

Our reply: As pointed out above by the reviewer, the triplet continuum for HAFC has a net spin $S = 1$, and has an equal contribution from the $S_z = 0, \pm 1$ sectors. If we understand correctly, then the reviewer’s comment might stem from the different terminology used in RIXS and INS community. By $\Delta S = 0$, we meant excitation within the singlet manifold, which is very different from the $S_z = 0$ sector of the triplet manifold accessed in INS. *We have added a note to discriminate $S = 0$ from $S_z = 0$ (of triplet manifold) excitations in the revised manuscript.* We agree with the referee that 2-magnon scattering in light scattering (e.g. optical Raman spectroscopy) is analogous to O K RIXS in respect to probing magnetic excitations, but note that optical Raman spectroscopy is limited to probing only Gamma, whereas O K RIXS is momentum dependent and probes a significant part of the BZ around

Gamma. In addition, RIXS has benefits from intermediate state effects that Raman measurements do not.

The reviewer wrote: To summarize, I endorse publishing this paper in Nature Communications, but ask the authors to fix some of the statements about neutron scattering and maybe tone down the claims about the novelty of the observation of “4 spinon” states.

Our reply: We again thank the reviewer for the positive appraisal of our work. We have made changes in our revised manuscript to clarify some of the statements. While we do feel the four spinons measured here are important, we have also toned down some of the claims where appropriate.

REVIEWERS' COMMENTS:

Reviewer #2 (Remarks to the Author):

I am satisfied with the author's reply to my and another referee's comments. I also believe that the text of the manuscript is improved and can be published in a current form. I do believe that the measurements of the four-spinon continuum by RIXS are important and, therefore, deserve a publication in Nat Comm.

Reviewer #3 (Remarks to the Author):

I find that the authors have given reasonable and thoughtful replies to the previous comments as well as to the adjustments to the manuscript. I agree that this is potentially an important contribution to the literature and have no hesitation to recommend publication of the current version in Nature Communications.

As an aside, as the authors inferred I was referring to ΔS_z in the discussion, and I was trying to follow what I assumed to be their notation. I was confused somewhat by the previous version referring to $\Delta S=1$ as "spin flip", since $\Delta S_{tot} = 1$ processes can also contain what are commonly referred to as non-spin flip (i.e. $\Delta S_z=0$) terms as well. Indeed INS will not show a signal for transitions with $S_{tot} = 0$ to $S_{tot} = 0$. The revised manuscript clarifies this.

Response to the Reviewers

The Reviewer #2 wrote: I am satisfied with the author's reply to my and another referee's comments. I also believe that the text of the manuscript is improved and can be published in a current form. I do believe that the measurements of the four-spinon continuum by RIXS are important and, therefore, deserve a publication in Nat Comm.

The Reviewer #3 wrote: I find that the authors have given reasonable and thoughtful replies to the previous comments as well as to the adjustments to the manuscript. I agree that this is potentially an important contribution to the literature and have no hesitation to recommend publication of the current version in Nature Communications.

As an aside, as the authors inferred I was referring to ΔS_z in the discussion, and I was trying to follow what I assumed to be their notation. I was confused somewhat by the previous version referring to $\Delta S=1$ as “spin flip”, since $\Delta S_{\text{tot}} = 1$ processes can also contain what are commonly referred to as non-spin flip (i.e. $\Delta S_z=0$) terms as well. Indeed INS will not show a signal for transitions with $S_{\text{tot}} = 0$ to $S_{\text{tot}} = 0$. The revised manuscript clarifies this.

Our reply: We would like to thank both reviewers for their time they have spent in examining our work and for their positive recommendations.